# Invasive Fungal Infections in Hospitalized Patients with COVID-19: A Non-Intensive Care Single-Centre Experience during the First Pandemic Waves

**DOI:** 10.3390/jof9010086

**Published:** 2023-01-06

**Authors:** Letizia Cattaneo, Antonio Riccardo Buonomo, Carmine Iacovazzo, Agnese Giaccone, Riccardo Scotto, Giulio Viceconte, Simona Mercinelli, Maria Vargas, Emanuela Roscetto, Francesco Cacciatore, Paola Salvatore, Maria Rosaria Catania, Riccardo Villari, Antonio Cittadini, Ivan Gentile

**Affiliations:** 1Department of Clinical Medicine and Surgery—Section of Infectious Diseases, University of Naples “Federico II”, 80131 Naples, Italy; 2Department of Neurosciences, University of Naples “Federico II”, 80131 Naples, Italy; 3Department of Molecular Medicine and Medical Biotechnology, University of Naples “Federico II”, 80131 Naples, Italy; 4Department of Translational Medical Sciences, University of Naples “Federico II”, 80131 Naples, Italy

**Keywords:** IFIs, COVID-19, pneumocystosis, invasive aspergillosis

## Abstract

Invasive fungal infections (IFIs) represent a severe complication of COVID-19, yet they are under-estimated. We conducted a retrospective analysis including all the COVID-19 patients admitted to the Infectious Diseases Unit of the Federico II University Hospital of Naples until the 1 July 2021. Among 409 patients, we reported seven cases of IFIs by *Candida* spp., seven of *Pneumocystis jirovecii* pneumonia, three of invasive pulmonary aspergillosis, and one of *Trichosporon asahii*. None of the cases presented underlying predisposing conditions, excluding one oncohematological patient treated with rituximab. Ten cases showed lymphopenia with high rates of CD4+ < 200/µL. All cases received high-dose steroid therapy (mean duration 33 days, mean cumulative dosage 1015 mg of prednisone equivalent), and seven cases had severe COVID-19 disease (OSCI ≥ 5) prior to IFI diagnosis. The cases showed a higher overall duration of hospitalization (63 vs 24 days) and higher mortality rate (23% vs. 7%) compared with the COVID-19 patients who did not developed IFIs. Cases showed a higher prevalence of high-dose steroid therapy and lymphopenia with CD4+ < 200/µL, primarily due to SARS-CoV-2 infection and not related to underlying comorbidities. IFIs strongly impact the overall length of hospitalization and mortality. Therefore, clinicians should maintain a high degree of suspicion of IFIs, especially in severe COVID-19 patients.

## 1. Introduction

Invasive fungal infections (IFIs) represent an important complication in patients hospitalized for COVID-19, especially severely ill ones during the middle and latter stages of the disease, being associated with a significant increase in mortality, morbidity and length of hospital stay [1,2,3,4,5,6,7,8].

The main fungal pathogens reported in severe COVID-19 patients are *Aspergillus* spp and *Candida* spp., followed by *Mucorales*, *Pneumocystis jirovecii* and other secondary IFIs usually diagnosed in severely immunocompromised patients (cryptococcosis, histoplasmosis, *Coccidioides* infections, *Fusarium* spp., and *Scedosporium* pulmonary infections) [9,10,11].

Risk factors and incidence of IFIs in COVID-19 are still challenging to define. IFIs incidence in the literature varies highly, primarily due to the heterogeneity of patients, surveillance protocols and parameters used for the definition of fungal infections. Actually, risk factors associated with these secondary infections include COVID-19-associated pathological mechanisms (i.e., epithelial barrier damage and immune system dysregulation leading to lymphopenia and decrease in CD4+T and CD8+T cells) or use of specific drugs during the disease (corticosteroids or immunomodulators) as well as healthcare-associated risk factors (i.e., widespread use of antibiotics predisposing to fungal colonization, parenteral nutrition, central venous catheters, admission to an intensive care unit (ICU), mechanical ventilation, and prolonged hospitalization) [7,9,12,13,14,15,16]. The aim of this study is to describe risk factors, incidence, and outcomes of invasive fungal superinfections in patients hospitalized for COVID-19 in a single non-intensive department.

## 2. Materials and Methods

We designed a retrospective observational study revising all the charts of patients with confirmed SARS-CoV-2 infection admitted to the Infectious Diseases (ID) Unit of the University Hospital “Federico II” of Naples from the 8 March 2020 to the 1 July 2021.

Disease severity was assessed using the 8-point WHO Ordinary Scale for Clinical Improvement (OSCI) [17]. In case of worsening requiring intensive therapies, the patients were moved to the ICU of the “Federico II” University Hospital, where the ID specialists kept collaborating with the intensivists for the management of these patients.

In our routinary clinical practice, we screened for invasive fungal infections (IFIs) all the patients with severe to critical COVID-19 disease and risk factors for IFIs and those showing a sudden clinical impairment with non-typical radiological or clinical features for COVID-19. WHO defined severe COVID-19 disease as an oxygen saturation <90% on room air and respiratory distress. Moreover, critical COVID-19 disease was defined by the criteria for acute respiratory distress syndrome (ARDS), sepsis, septic shock, or the need for life-sustaining therapies such as mechanical ventilation (invasive or non-invasive) or vasopressor therapy [18]. Risk factors for IFIs included: prolonged antimicrobial therapies, prolonged steroid therapy, central venous catheters, prolonged parenteral nutrition, neutropenia, underlying comorbidities such as solid or hematologic malignancies, acquired or inherited immunodeficiencies, recent HSCT or solid organ transplant. In all these patients, multiple sets of blood cultures were taken; (1,3)-beta-D glucan (BDG) and *Aspergillus* galactomannan (GM) were tested on serum using the Fungitell^®^ Assay (Associates of Cape Cod) for the former and Platelia Aspergillus (Bio-Rad) for the latter. Bronchoscopy and bronchoalveolar lavage (BAL) were performed upon high clinical suspicion of IFIs (i.e., compatible radiological images and/or sudden worsening of respiratory function non-justifiable with the clinical course of COVID-19 disease or bacterial superinfections) after acquiring written informed consent to the procedure. Cultures for bacteria and fungi with microscopic analysis, immunofluorescence assay (IFA) for *P. jirovecii* with MONOFLUID™ KIT (Axis-Shield Diagnostics Limited) and galactomannan test were performed on the bronchoalveolar lavage fluid (BALF). A HRCT scan of the thorax was performed in all the patients at admission (except in pregnant women, for whom it was replaced by chest ultrasound) and eventually repeated upon medical judgment.

We identified three major groups: patients who developed candidemia, patients who developed *Pneumocystis jirovecii* pneumonia (PJP), and patients who developed invasive pulmonary aspergillosis (IPA).

The IFIs were classified as “proven”, “probable”, or “possible” based on the Consensus definition from EORTC/MSGERC [19].

## 3. Results

We enrolled 409 patients with a very heterogeneous severity of COVID-19 at enrollment; in fact, it ranged from asymptomatic or paucisymptomatic infection to severe disease needing non-invasive ventilation (NIV) or high-flow oxygen therapy. Eighty-five patients had critical SARS-CoV-2-related pneumonia and respiratory failure requiring ICU admission during the hospital stay. The overall mortality rate was 7.6% (31/409). Eighteen invasive fungal infections were diagnosed in 13 patients: seven cases of *Candida* spp. candidemia (three cases of *C. albicans*, three cases of *C. parapsilosis*, one case of *C. glabrata*), seven cases of PJP, three cases of IPA, one case of *Trichosporon asahii* fungemia.

All but one *Candida* spp. bloodstream infections were central line-associated bloodstream infections and, in two cases, developed immediately after discharge from the ICU ward. In the other cases, we supposed that the worsening of the clinical condition was due to the IFI. Actually, immediately after ICU admission, the IFI diagnosis was made. Echinocandin was the chosen treatment in all these conditions, which lasted 14 days or more (up to 10 days after central venous catheter removal) (Table 1).

Among PJPs, all but one developed before ICU admission; in the single case in which IFI developed after ICU admission, the respiratory failure that led to orotracheal intubation was due to pneumocystosis. Trimethoprim/Sulfamethoxazole was the drug of choice in this setting for 21 days, together with steroidal treatment (Table 1).

Among IPAs, in two cases it was diagnosed the day after the orotracheal intubation; in the third case, it developed during hospitalization in the ID ward. Invasive pulmonary aspergillosis has been treated with isavuconazole for at least three months.

Finally, *Trichosporon asahii* fungemia was treated with intravenous isavuconazole for 14 days as it developed together with *Candida parapsilosis* fungemia (Table 1).

### 3.1. Characteristics of COVID-19 Patients Who Developed IFIs

In total, 13 patients out of 409 (3%) developed one or more IFIs. In detail, 6 patients developed candidemia; 7 patients developed PJP; 3 patients developed IPA. In addition, 3 patients developed more than one IFI (Table 1). Among the 13 patients who developed IFIs, 9 were males (69%), 4 were females (31%), and their median age was 69 years (IQR: 53.5–75). Additionally, 2 patients were pregnant: they first tested positive for SARS-COV-2 during the third and second trimesters of a previously uncomplicated pregnancy. Table 2 lists the main features of these patients.

All the patients who developed IFIs required supplemental oxygen therapy during the hospital stay (maximum WHO-OSCI score ≥ 4); 7 patients (54%) presented severe COVID-19 (WHO-OSCI score ≥ 5) at IFI diagnosis. 

All these patients received steroids, according to the standard of care for COVID-19. At IFIs diagnosis, the mean steroid duration had been 38.5 days (median, 31.5; IQR 15–58.25), and the mean cumulative prednisone-equivalent steroid dose was 1028.5 mg (median, 630 mg; IQR 447–1433). None of the patients who developed IFIs received COVID-19-specific immunotherapy such as tocilizumab. During the hospitalization, nine patients (69%) developed lymphocytopenia. Eight patients (88%) performed a lymphocytes subset analysis: among them, five (62.5%) patients showed CD4+ count <200 cells/µL.

The median time frame from the COVID-19 onset to the IFI diagnosis was 39.5 days (IQR 13–65). Six patients (46%) were admitted to ICU prior to the diagnosis of IFIs. The mean duration of ICU stay prior to IFI diagnosis was 12 days (median, 5; IQR 2–21.5).

Three patients died (23%) (Table 2).

According to EORTC/MSGERC Consensus definitions of IFIs, only one patient diagnosed with PJP infection, presented an underlying comorbidity consistent with an increased risk of fungal infections, namely a hematologic malignancy (non-Hodgkin lymphoma, NHL) on treatment with rituximab [19]. None of the patients who developed IFIs tested positive for HIV or presented neutropenia during the hospital stay.

### 3.2. Characteristics of Each Diagnosed IFI

The characteristics of each group are summarized in Table 3.

In total, 67% of the patients in both *Candida* and IPA groups needed intensive treatment and ICU admission before IFIs diagnosis. In one case a candidemia was diagnosed when the patient had already been moved to the ID ward; in the other cases, the worsening of clinical conditions leading to ICU admission was soon after related to IFI’s development.

None of the patients who developed IFIs received any immunomodulant therapy for COVID-19.

A major difference could be observed in the mean cumulative prednisone-equivalent steroid dose administered to the patients in the PJP and IPA groups, calculated from the COVID-19 onset until the day of the IFI diagnosis. The steroid therapy resulted prolonged and at a higher dose in the PJP group vs IPA group (1301 mg prednisone equivalent vs. 1026 mg).

Moreover, the *Candida* and the IPA groups showed a remarkably high mortality rate (33%), while in the PJP group no death was reported.

## 4. Discussion

### 4.1. Incidence

The incidence of IFIs in our cohort was 3%, lower than the incidence reported by Casalini et al. in one of the latest reviews on this topic, namely 6.9% [15]. However, we can hardly compare this incidence with that reported in the literature since a wide variability exists in publications [9].

In our cohort, six patients developed candidemia (1%), seven patients developed PJP (1.7%), and three patients developed IPA (0.7%).

The incidence of IPA in our cohort is in contrast with the currently available literature, which reports pulmonary aspergillosis as the most prevalent IFI in COVID-19 patients, with an average incidence of 3.9% and an incidence in observational studies ranging from 0.1% to 57% [15,20]. This can be partially explained by taking into account that the majority of the studies on IFIs in COVID-19, particularly on COVID-19-associated pulmonary aspergillosis (CAPA), were conducted in ICU settings, where mechanical ventilation greatly increases the risk of fungal colonization potentially leading to invasive pulmonary aspergillosis [3,15,21]. However, further studies are needed to investigate if the ICU setting provides independent risk factors for the development of pulmonary aspergillosis other than the severity of COVID-19 disease since the OSCI score of our patients who developed IPA was 4, consistent with higher severity.

In contrast, the incidence of candidemia in our study is consistent with that reported in the literature, which ranges from 0.03 to 14% [3].

Conversely, the incidence of PJP among our patients was significantly higher than what the literature reports: Casalini et al., in their review, found that only 20 cases of *Pneumocystis jirovecii* pneumonia (PJP) were reported among 4099 cases of IFIs in 58,784 COVID-19 patients in 168 studies (incidence 0.03%) [15]; Baddley et al. described only 8 cases of PJP, but all patients except one presented an underlying immunosuppressive condition (six patients were HIV positive and one had received kidney transplantation) [9]. Therefore, it can be speculated that PJP is frequently neglected in COVID-19 patients without an underlying predisposing condition. In fact, as already mentioned, in our cohort, all but one patient had no pre-existing immunosuppressive conditions or other risk factors for PJP development upon COVID-19 diagnosis. Moreover, all our patients developed PJP after the clinical resolution of COVID-19 pneumonia. This complication might have been undiagnosed if the patients were discharged or lost to follow-up. Furthermore, BDG was not detectable in all the serum samples taken from the patients with proven PJP, supporting the hypothesis of a lung-localized PJP instead of the invasive infection with fungemia which is usually observed in AIDS patients (recent data on the diagnostic accuracy of BDG for *P. jirovecii* pneumonia report a sensitivity of 94% in HIV-patients versus 86% in non-HIV patients with a comparable specificity of 83%) [16,22].

### 4.2. Time of Onset

In our study, the timeframe from the COVID-19 onset to the IFIs diagnosis (median, 39.5 days; IQR 13–65) and the duration of hospitalization at IFIs diagnosis (median, 16.5 days; IQR 7.75–40.75) reflect, in accordance with the literature, that the risk of IFIs is higher in the late course of COVID-19 disease when the lung damage and the alteration of the immune system predominate and the likelihood of overlapping healthcare-related infections arises.

Unlike other studies, where PJP usually occurs before CAPA, [23] in our cohort, pulmonary aspergillosis occurred first (median delay, 3.5 days; IQR 2–5), followed by *Candida* spp. bloodstream infection (median delay, 10.25 days; IQR 2–28) and PJP as belated diagnosis (median delay, 30 days; IQR 15–45).

The time of onset of fungemia in our cohort was similar to the one reported in the literature, in accordance with the pooled mean duration of ICU stay before the onset of candidemia found by a recent meta-analysis (12.9 days) [15]. This data on candidemia support the hypothesis that this infection could be related to prolonged hospitalization, critical condition and/or ICU admission rather than COVID-19-related risk factors.

### 4.3. COVID-19 Related Risk Factors

As already highlighted in the literature, in our cohort high-dose corticosteroid administration appears to be the major risk factor for the development of IPA and PJP in COVID-19 patients [9,11,15,16]. In fact, both the mean duration of steroid therapy and the cumulative prednisone-equivalent steroid dose (from the COVID-19 onset until the day of the IFIs diagnosis) were high in the little group of patients who developed IFIs. Moreover, lymphopenia with low CD4+ cell count represented a consistent finding in patients diagnosed with PJP or IPA.

In our study, in agreement with Casalini et al., [15] fungemia and IPA frequently occurred in patients with a very severe course of COVID-19, as 67% of our patients in both *Candida* and IPA groups needed ICU admission before the IFIs onset. Even according to the OSCI scale, 83% and 67% of the patients who developed respectively candidemia or IPA had a severe COVID-19 disease defined by an OSCI score ≥5, while only 29% of patients developing PJP had a critical disease. Therefore, an interesting finding of our study is that these infections also occurred in patients not previously admitted to ICU or with a very short ICU stay prior to IFIs diagnosis.

Finally, we underline that most patients did not present classical predisposing conditions for IFIs development. For this reason, we suggest that COVID-19 itself may represent an independent risk factor for developing fungal infections. This may be mediated by several mechanisms: COVID-19-induced lymphopenia, virus-induced alveolar damage, and impaired immune response following prolonged steroid therapy [24].

### 4.4. Outcomes

In our study, the overall mortality rate in the IFIs group was 23%, lower than the mortality rate found by White et al. (38.5%) [8]. In particular, mortality in our *Candida* group (33%) was much lower than that reported by other studies (where it ranged between 40–75%) [9,15]. Even the mortality of the IPA group in our cohort was lower than the one described in the literature (33% vs. 48–56%) [3,9,15]. Bretagne et al., in their study, reported a mortality rate of 29.4% for CA-PJP [21], while in our population, no death was recorded in this group. This discrepancy may be due primarily to the different hospital settings where the study took place (non-ICU vs. ICU) but also to the high level of suspicion for IFIs that characterized our team in the management of COVID-19 patients due to previous experience, which led to early diagnosis and treatment and consequently better outcome [10,11,24].

Nonetheless, our study confirms that the development of IFIs is associated with a worse outcome, particularly in *Candida* and IPA groups, where the mortality rate was 33%.

## 5. Conclusions

Invasive fungal infections are not uncommon in patients with COVID-19 due to the interplay of pre-existing predisposing conditions, health-care-associated risk factors, and COVID-19-associated pathological mechanisms. In many cases, risk factors for IFIs and appropriate diagnostic strategies have been challenging to define, which has led to a wide variability of IFIs incidence and reported outcomes. Although CAPA and candidemia raised the most concerns, our observational study suggests that invasive mycoses typically observed in highly immunocompromised patients can also be found in COVID-19 patients. We, therefore, emphasize the importance of a high degree of suspicion and strive for an early diagnosis, as the occurrence of invasive fungal infections in critically ill patients may have a serious impact on morbidity and mortality. Nonetheless, due to the numerous limits of our study—namely the small cohort, as a single-centre study, and the retrospective observational design—we did not succeed in reaching statistically significant data. Further prospective—and preferably multi-center—studies are needed to better define the burden of invasive fungal infections in COVID-19 patients.

Patients diagnosed with IFI were treated according to the latest IDSA guidelines [25,26,27].

## Figures and Tables

**Table 1 jof-09-00086-t001:** Invasive fungal infections (IFIs) in patients with SARS-CoV-2 infection admitted to the Infectious Diseases Unit (IDU) of the University Hospital “Federico II” of Naples from the 8 March 2020 to the 1 July 2021.

Patient	Age, Years	Gender	Lenght of Stay, Days	Clinical Severity (8-Point WHO-OS) at IFI Diagnosis	Fungus	Diagnosis	Days Since Hospital Admission	ICU Admission Prior to IFI	Lenght of ICU Stay Prior to IFI, Days	Days Since COVID-19 Onset	Steroid Duration Prior to IFI, Days	Cumulative Steroid Dose (Prednisone Equivalent) Prior to IFI, mg	Lymphocytes Count, n/µL	CD4+ Count, n/µL	Outcome
1	75	M	121	4	*Candida albicans*	Proven	36	No	n/a	37	37	327	430	192	Recovery
				4	*Candida parapsilosis*		94	Yes	2	95	95	447	870	356	
				4	*Trichosporon asahii*										
2	75	M	31	5	*Candida albicans*	Proven	15	No	n/a	13	n/a	n/a	540	n/a	Death
3	86	F	54	5	*Candida glabrata*	Proven	46	Yes	2	52	53	1433	3920	n/a	Death
4	69	M	38	5	*Candida albicans*	Proven	8	No	n/a	11	10	586	800	n/a	Recovery
5	38	F	22	5	*Candida parapsilosis*	Proven	13	Yes	9	11	n/a	n/a	3660	n/a	Recovery
6	63	M	77	6	*Aspergillus* spp.	Probable	7	Yes	2	13	13	440	900	648	Death
7	79	M	41	4	*Aspergillus fumigatus*	Proven	5	No	n/a	65	60	1785	560	140	Recovery
				4	*Pneumocystis jirovecii*										
8	63	F	78	4	*Pneumocystis jirovecii*	Proven	18	No	n/a	120	15	630	240	93	Recovery
9	52	M	18	4	*Pneumocystis jirovecii*	Proven	39	No	n/a	40	32	963	1260	895	Recovery
10	72	M	24	4	*Pneumocystis jirovecii*	Possible	15	No	n/a	26	20	475	2200	1012	Recovery
11	55	M	22	4	*Pneumocystis jirovecii*	Proven	3	No	n/a	45	15	400	260	62	Recovery
12	69	M	56	5	*Pneumocystis jirovecii*	Possible	28	Yes	15	39	25	1150	590	141	Recovery
13	31	F	62	6	*Aspergillus fumigatus*	Proven	8	Yes	5	11	8	853	860	n/a	Recovery
				6	*Candida parapsilosis*		31		28	34	31	n/a	720		
				6	*Pneumocystis jirovecii*		48		45	51	48	3707	380		

**Table 2 jof-09-00086-t002:** Summary of the characteristics of the 13 COVID-19 patients who developed IFIs.

Characteristics	Results
Age (years), mean (median, IQR)	63.6 (69; 53.5–75)
Male sex, *n* (%)	9 (69)
Risk factors for IFIs, *n* (%)	2 (15)
Hematologic malignancy	1 (8)
HIV/AIDS	0 (0)
Neutropenia	0 (0)
Immunotherapy	1 (8)
Supplemental oxygen therapy, *n* (%)	13 (100)
WHO-OSCI score ≥5, *n* (%)	8 (62)
ICU admission prior to IFI, *n* (%)	6 (46)
Length of ICU stay prior to IFI (days), mean (median, IQR)	12 (5; 2–21.5)
Length of hospital stay prior to IFI (days), mean (median, IQR)	28.5 (16.5; 7.75–40.75)
Time since COVID-19 onset (days), mean (median, IQR)	45.7 (39.5; 13–65)
Cumulative prednisone-equivalent steroid dose prior to IFI (mg), mean (median, IQR)	1028.5 (630; 447–1433)
Duration of steroid therapy prior to IFI (days), mean (median, IQR)	38.5 (31.5; 15–58.25)
Lymphocytopenia (<1000 cells/µL), *n* (%)	9 (69)
CD4+ count < 200 cells/µL	5/8 (63)
Overall length of hospital stay (days), mean (median, IQR)	49.5 (41; 23–69.5)
Overall mortality, *n* (%)	3 (23)

**Table 3 jof-09-00086-t003:** Characteristics of COVID-19 patients who developed IFIs clustered by fungus genera.

Invasive Candidiasis	N = 6
ICU admission *, *n* (%)	4 (67)
Severe COVID-19 **, *n* (%)	5 (83)
Tocilizumab, *n* (%)	0 (0)
Length of Stay (days), mean	55
Overall mortality, *n* (%)	2 (33)
*Pneumocystis jirovecii* Pneumonia (PJP)	N = 7
ICU admission *, *n* (%)	2 (29)
Severe COVID-19 **, *n* (%)	2 (29)
Hematologic malignancy, *n* (%)	1 (14)
Immunotherapy, *n* (%)	1 (14)
Tocilizumab, *n* (%)	0 (0)
Cumulative steroid dosage (prednisone equivalent, mg) *, mean	1301
Lymphopenia, *n* (%)	5 (71)
CD4+ < 200 n/µL, *n* (%)	4/6 (67)
HIV infection, *n* (%)	0 (0)
AIDS, *n* (%)	0 (0)
Length of Stay (days), mean	43
Overall mortality, *n* (%)	0 (0)
Post COVID-19 Invasive Pulmonary Aspergillosis (IPA)	N = 3
ICU admission *, *n* (%)	2 (67)
Severe COVID-19 **, *n* (%)	2 (67)
Hematologic malignancy, *n* (%)	0 (0)
Immunotherapy, *n* (%)	0 (0)
Tocilizumab, *n* (%)	0 (0)
Cumulative steroid dosage (prednisone equivalent, mg) *, mean	1026
Lymphopenia, *n* (%)	3 (100)
CD4+ <200 n/µL, *n* (%)	1/2 (50)
HIV infection, *n* (%)	0 (0)
AIDS, *n* (%)	0 (0)
Length of Stay (days), mean	60
Overall mortality, *n* (%)	1 (33)

* Prior to IFI diagnosis for the IFIs groups or at hospital discharge for the control population. ** Severe disease defined by 5 to 7 points (8-point WHO-OSCI) at the time of IFI diagnosis for the IFIs groups or at any stage for the control population.

## Data Availability

The data that support the findings of this study are available on request from the corresponding author, A.R.B. The data are not publicly available due to privacy restrictions.

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
