# Peer review of "Invasive Fungal Infections in Hospitalized Patients with COVID-19: A Non-Intensive Care Single-Centre Experience during the First Pandemic Waves"

_jof, 2023, doi:10.3390/jof9010086_

Round 1

Reviewer 1 Report

In this manuscript the authors describe a single centre experience in COVID-19 associated fungal infections from the first pandemic wave/s. the topic interesting as it is focused on non-ICU patients, as compared to most of the publications. Nevertheless, certain corrections need to be done before I may recommend it for publication.

MAJOR COMMENTS

- There has to be some logic in the case-control comparison. In your publication I think this does not exist. If I understand correctly, you compare the cases with a certain IFI to all those without that specific IFI. This does not make sense, you are including in controls patients with IFI and with no IFI.

- There is no paragraph for limitations.

- In my opinion, the asset of your publication is precisely that you are focusing on patients with no ICU admission. Thus, you cannot compare your data in the discussion with those from patients in ICU. Or at least not with the current phrasing. I suggest to compare your sample to non-ICU patients, when possible. When comparing it to ICU patients, if you observe that all your parameters, incidences, mortality, etc are "better", you can even draw the hypothesis that this can be related to the fact that your patients were not admitted in ICU. Or any other hypothesis you might speculate.

- "we preferred the former definition of Invasive Pulmonary Aspergillosis (IPA) for our COVID-19 patients who developed IFI by Aspergillus spp.". Which is? There is not reference not explanation. I would suggest to use the EORTC/MSG one, or even try the ECMM/ISHAM for CAPA, excluding the entry criterion of ICU admission. Or even both. Such definition is not specific for non-ICU patients, so definitely you have to get your own definition.

- Which were the sites of infection? How were your patients diagnosed? How were they treated? All but one was completely healthy before COVID-19? Not even a diabetes mellitus?

- Six patients were admitted in the ICU. Please, explain when and why they can be considered for the IFI as non-ICU.

MINOR COMMENTS

- Line 39: "Mucor" is a genus, "Mucorales" is the term you should use, the order.

- Do not smash all the references at the end of the respective paragraph, please, include them immediately after the sentence/s it makes reference to.

- Your data can be already considered "historical" data. Thus, I would suggest to make such statement clear, including in the title. Thus, I would suggest to refer to "historical" literature too.

- Include previous reports on incidences. For CAPA I recommend Salmanton-García et al. Emerging Infectious Diseases 2021, where incidences in different settings are provided.

- I would suggest to give more relevance to the Trichosporon case, are there are very few cases reported.

- In Methods you are making reference to previous publications, please, include all the references. Also include appropriately all the definitions you are using within your manuscript.

- Please, explain how are you analysing your data (statistical analysis) in a last paragraph in methods.

- Always provide n and % together.

- Improve the image quality of your tables.

- You compare data between "cases" and "controls". Which are the statistical results?

- Please, compare your results to actual reports, not to reviews, like ref #15.

Author Response

Response to review report 1

In this manuscript the authors describe a single centre

experience in COVID-19 associated fungal infections from the

first pandemic wave/s. the topic interesting as it is focused on

non-ICU patients, as compared to most of the publications.

Nevertheless, certain corrections need to be done before I may

recommend it for publication.

MAJOR COMMENTS

- There has to be some logic in the case-control comparison. In

your publication I think this does not exist. If I understand

correctly, you compare the cases with a certain IFI to all those

without that specific IFI. This does not make sense, you are

including in controls patients with IFI and with no IFI.

R: We strongly agree with the referee and we thank him/her for the comment.

Therefore, we modified methods and results sections. Moreover, we modified the table, and we removed the column in which were reported data referring to non IFI group.

- There is no paragraph for limitations.

R: We thank the reviewer for the comment. Actually, we already included limitations of the study in the conclusion paragraph (pag 7, line 332-336).

- In my opinion, the asset of your publication is precisely that you

are focusing on patients with no ICU admission. Thus, you

cannot compare your data in the discussion with those from

patients in ICU. Or at least not with the current phrasing. I

suggest to compare your sample to non-ICU patients, when

possible. When comparing it to ICU patients, if you observe that

all your parameters, incidences, mortality, etc are "better", you

can even draw the hypothesis that this can be related to the fact

that your patients were not admitted in ICU. Or any other

hypothesis you might speculate.

R: We agree with the reviewer’s comment and we already stated in the discussion that is difficult to compare two populations with different severity of the underlining.

- "we preferred the former definition of Invasive Pulmonary

Aspergillosis (IPA) for our COVID-19 patients who developed IFI

by Aspergillus spp.". Which is? There is not reference not

explanation. I would suggest to use the EORTC/MSG one, or

even try the ECMM/ISHAM for CAPA, excluding the entry

criterion of ICU admission. Or even both. Such definition is not

specific for non-ICU patients, so definitely you have to get your

own definition.

R: We thank the reviewer for the comment and we agree with his/her claim. We modified the manuscript and we clarified what criteria we adopted.

- Which were the sites of infection? How were your patients

diagnosed? How were they treated?

R: Dear Reviewer, in the manuscript (page 5, line 208-209) you can already find the following sentence:“In our cohort, 6 patients developed candidemia (1%), 7 patients developed PJP (1.7%), 3 patients developed IPA (0.7%).” We modified the manuscript and we clarify how we managed different IFIs. (page 3, line 132-146)

- All but one was completely healthy before COVID-19? Not even a diabetes mellitus?

R: We thank the reviewer for the comment. We checked again the database and we can confirm that all but one patient had no risk factor for IFI’s development according to IFI guidelines.

- Six patients were admitted in the ICU. Please, explain when

and why they can be considered for the IFI as non-ICU.

R: We thank the reviewer for the comment and we better explained this result.

Now, we explained that: “67% of the patients in both Candida and IPA group needed intensive treatment and ICU admission before IFIs onset. In one case a candidemia has been diagnosed when the patient was already hospitalized in internal ward, in the other cases the worsening of clinical condition leading to ICU admission, was soon after related to IFI’s development.”(page 4, line 185-188)

MINOR COMMENTS

- Line 39: "Mucor" is a genus, "Mucorales" is the term you should

use, the order.

R: Done.

- Do not smash all the references at the end of the respective

paragraph, please, include them immediately after the

sentence/s it makes reference to.

R: Done.

- Your data can be already considered "historical" data. Thus, I

would suggest to make such statement clear, including in the

title. Thus, I would suggest to refer to "historical" literature too.

R: Done.

- Include previous reports on incidences. For CAPA I recommend

Salmanton-García et al. Emerging Infectious Diseases 2021,

where incidences in different settings are provided.

R: Done.

- I would suggest to give more relevance to the Trichosporon

case, are there are very few cases reported.

R: We thank the reviewer for this suggestion. We modified the manuscript in order to better explain how we managed this case. (pag 3, line 146-147).

- In Methods you are making reference to previous publications,

please, include all the references. Also include appropriately all

the definitions you are using within your manuscript.

R: Done.

- Please, explain how are you analysing your data (statistical

analysis) in a last paragraph in methods.

R: As we explain in the conclusion paragraph “Nonetheless, due the numerous limits of our study – namely the small cohort, as a single-centre study, and the retrospective observational design – we did not succeed in reaching statistically significant data.” So we did not do a statysticak analysis.

- Always provide n and % together.

R: Done.

- Improve the image quality of your tables.

R: Done.

- You compare data between "cases" and "controls". Which are

the statistical results?

R: We modified the manuscript because of lacking of statistical significance.

- Please, compare your results to actual reports, not to reviews,

like ref #15.

R: Dear Reviewer, unfortunately literature on IFIs in COVID-19 is still poor so we could not always compare our data to actual reports. This was the primum movens of our study and that’s why we stress the importance of collecting and publishing data on IFIs in COVID-19.

Reviewer 2 Report

In this manuscript, the authors conducted an analysis of the occurence as well as outcome of the invasive fungal infections in patients with COVID 19 infection and its corelation to associated risk factors. Some of the findings are in contrast to what has been previously published but that is well explained and justified in their discussion. 

Line 26: Add space before “Cases”

Line 28: Remove “on” before overall

Line 45: Can be rephrased as “Parameters used for the definition of fungal infections”

Line 83-83: Needs rephrasing

Line 90: Format “consensus”

Line 97: What does author want to imply by “respectively IPA”?

Line 117-119: inconsistent formatting

Line 125-126: Needs rephrasing

Line 130: Needs more detail

Line 154-156: Unclear what the author wants to convey

Well written discussion.

Author Response

Response to review report 2

In this manuscript, the authors conducted an analysis of the occurence as well as outcome of the invasive fungal infections in patients with COVID 19 infection and its corelation to associated risk factors. Some of the findings are in contrast to what has been previously published but that is well explained and justified in their discussion.

Line 26: Add space before “Cases”

R: Done.

Line 28: Remove “on” before overall

R: Done.

Line 45: Can be rephrased as “Parameters used for the definition of fungal infections”

R: Done.

Line 83-83: Needs rephrasing

R: Done.

Line 90: Format “consensus”

R: Done.

Line 97: What does author want to imply by “respectively IPA”?

R: we thank the reviewer for the comment. We modified the manuscript and we remove this sentence.

Line 117-119: inconsistent formatting

R: Done.

Line 125-126: Needs rephrasing

R: We rephrase the sentence.

Line 130: Needs more detail

R: We clarify. Now it reads:

“Eighteen invasive fungal infections were diagnosed in 13 patients: 7 cases of Candida spp candidemia (3 cases of C. albicans, 3 cases of C. parapsilosis, 1 case of C. glabrata), 7 cases of PJP, 3 cases of IPA, 1 case of Trichosporon asahii fungemia.

All but one Candida spp. bloodstream infections were Central line associated bloodstream  infections and in two cases developed immediately after the discharge from ICU ward. In the other cases we supposed that the worsening of the clinical condition was due to the IFI. Actually, immediately after ICU admission, IFI diagnosis has been made. Echinocandin was the chosen treatment in all these conditions and it last 14 days or more (up to ten days after central vein catheter removal). (Table 1)

Among PJP, all but one did not develop after ICU admission; in the single case in which IFI developed after ICU admission, respiratory failure that led to orotracheal intubation was pneumocystosis. Trimetoprim/Sulfametoxazole was the drug of choice in this setting for 21 days together with steroidal treatment. (Table 1)

Among IPAs, in two cases it was diagnosed the day after the orotracheal intubation, in the other case it developed during hospitalization in infectious diseases ward. Invasive pulmonary aspergillosis has been treated with isavuconazole for at least three months.

Finally, Trichosporon asahii fungemia has been treated with intravenous isavuconazole for 14 days as it developed together with a Candida parapsilosis fungemia. (Table 1)”

Line 154-156: Unclear what the author wants to convey

R: We rephrase the sentence, now it reads:

“The steroid therapy resulted prolonged and at higher dose in the PJP group vs IPA group (1301 mg prednisone equivalent vs 1026 mg).

Moreover, the Candida and the IPA groups showed a remarkably high mortality rate (33%), while in the PJP group no death was reported.” (page 5, line 197-201)

Well written discussion.

Reviewer 3 Report

Cattaneo et al. conducted a retrospective study titled “INVASIVE FUNGAL INFECTIONS IN HOSPITALIZED  PATIENTS WITH COVID-19: A NON -INTENSIVE CARE  SINGLE CENTRE EXPERIENCE”. The authors present an interesting manuscript in which they investigate fungal co-infections in Covid-19 patients. Although the topic is of utmost importance, the manuscript has the following shortcomings: 

-       Although the authors state in their title and introduction that the present work investigates fungal co-infections in non-ICU patients, almost 50% of confirmed infections were diagnosed after ICU admission. Hence, the reported frequencies are somewhat misleading. The prevalence of non-ICU and ICU infections cannot be pooled and compared to other literature. Please revise the body of the manuscript or the title/objectives accordingly.

-       The authors refer to the consensus definition from EORTC/MSGERC according to the 2020 ECMM/ISHAM 90 consensus criteria for COVID-associated pulmonary aspergillosis (CAPA). Please, give an overview of exact criteria that were chosen for the definitions of each fungal infection in the present work. The cited criteria present guidance in clinically diagnosing fungal diseases when treating patients. However, when assessing data retrospectively, it may be difficult to apply these guidelines.

Please include the testing frequency when referring to co-infection incidences. I.e., how many patients were microbiologically sampled, and how many of which yielded positive results (fungal infections)?

Author Response

Review report 3

Cattaneo et al. conducted a retrospective study titled “INVASIVE

FUNGAL INFECTIONS IN HOSPITALIZED PATIENTS WITH

COVID-19: A NON -INTENSIVE CARE SINGLE CENTRE

EXPERIENCE”. The authors present an interesting manuscript

in which they investigate fungal co-infections in Covid-19

patients. Although the topic is of utmost importance, the

manuscript has the following shortcomings:

- Although the authors state in their title and

introduction that the present work investigates fungal co-

infections in non-ICU patients, almost 50% of confirmed

infections were diagnosed after ICU admission. Hence,

the reported frequencies are somewhat misleading. The

prevalence of non-ICU and ICU infections cannot be

pooled and compared to other literature. Please revise

the body of the manuscript or the title/objectives

accordingly.

R: We thank the reviewer for this comment. We modified the manuscript in order to better explain our results. Now it reads:

“Eighteen invasive fungal infections were diagnosed in 13 patients: 7 cases of Candida spp candidemia (3 cases of C. albicans, 3 cases of C. parapsilosis, 1 case of C. glabrata), 7 cases of PJP, 3 cases of IPA, 1 case of Trichosporon asahii fungemia.

All but one Candida spp. bloodstream infections were Central line associated bloodstream  infections and in two cases developed immediately after the discharge from ICU ward. In the other cases we supposed that the worsening of the clinical condition was due to the IFI. Actually, immediately after ICU admission, IFI diagnosis has been made. Echinocandin was the chosen treatment in all these conditions and it last 14 days or more (up to ten days after central vein catheter removal). (Table 1)

Among PJP, all but one did not develop after ICU admission; in the single case in which IFI developed after ICU admission, respiratory failure that led to orotracheal intubation was pneumocystosis. Trimetoprim/Sulfametoxazole was the drug of choice in this setting for 21 days together with steroidal treatment. (Table 1)

Among IPAs, in two cases it was diagnosed the day after the orotracheal intubation, in the other case it developed during hospitalization in infectious diseases ward. Invasive pulmonary aspergillosis has been treated with isavuconazole for at least three months.

Finally, Trichosporon asahii fungemia has been treated with intravenous isavuconazole for 14 days as it developed together with a Candida parapsilosis fungemia. (Table 1)” (pag 3, line 134-148).

Now we also explain that:

“67% of the patients in both Candida and IPA group needed intensive treatment and ICU admission before IFIs onset. In one case a candidemia has been diagnosed when the patient was already hospitalized in internal ward, in the other cases the worsening of clinical condition leading to ICU admission, was soon after related to IFI’s development.” (Page 4, line 185-188)

- The authors refer to the consensus definition from

EORTC/MSGERC according to the 2020 ECMM/ISHAM

90 consensus criteria for COVID-associated pulmonary

aspergillosis (CAPA). Please, give an overview of exact

criteria that were chosen for the definitions of each fungal

infection in the present work. The cited criteria present

guidance in clinically diagnosing fungal diseases when treating patients.

However, when assessing data

retrospectively, it may be difficult to apply these

guidelines.

R: We thank the reviewer for the comment. We modified the manuscript so that it would be more clear now (pag 2, line 89-91). Fortunately, we followed the IFIs patients of our cohort from the suspect, to the diagnosis till the outcome because we are part of the ID specialist team that makes consultation in all the different unit or ward of our University Hospital, as we explain in Methods paragraph (pag 2, 63-65), so it was easy to use the same diagnostic criteria according to the consensus definition from EORTC/MSGERC.

Please include the testing frequency when referring to co-

infection incidences. I.e., how many patients were

microbiologically sampled, and how many of which yielded

positive results (fungal infections)?

R: We thank the reviewer for the comment. Unfortunately, we did not have this kind of information due to retrospective nature of the study we are performing. However in method section we have clarified this limitation.

"In our routine clinical practice, we screened for invasive fungal infections (IFIs) all the patients with severe to critical COVID-19 disease and risk factors for IFIs and those showing a sudden clinical impairment with non-typical radiological or clinical features for COVID-19. " (page 2, line 66-69).

Reviewer 4 Report

The authors investigated the incidence of IFIs in a population of COVID-19 patients. Although this is a widely debated argument, it is still very interesting.

The study is great and well conducted. English language need only minor spell check. Only few comments (following) for the methods section.

Moreover, I have the following concerns:

-          Line 45: Please avoid words’ repetition (use a synonym of “definition” or rephrase the sentence)

-          Lines 117-119: Please check the font and the format

-          It is not clear what was the diagnostic methods for PJP? Did you use PCR, IFA alone, or histology? Please specify since it is a crucial point.

-          I think that it would be great adding few lines about treatment of patients you described (also, if there are some differences between proven, probable, and possible in terms of therapy). Moreover, it would be useful to know how you managed those IFIs.

-          I suggest reading and discuss these published papers that treat the same subject: 10.3390/idr14030041, 10.3892/br.2022.1517

Kind regards

Author Response

Response to review report 4

The authors investigated the incidence of IFIs in a population of COVID-19 patients. Although this is a widely debated argument, it is still very interesting.

The study is great and well conducted. English language need only minor spell check. Only few comments (following) for the methods section.

Moreover, I have the following concerns:

Line 45: Please avoid words’ repetition (use a synonym of “definition” or rephrase the sentence).

R: Done.

Lines 117-119: Please check the font and the format.

R: Done.

It is not clear what was the diagnostic methods for PJP? Did you use PCR, IFA alone, or histology? Please specify since it is a crucial point.    

R: The diagnostic methods for PJP was IFA alone.

I think that it would be great adding few lines about treatment of patients you described (also, if there are some differences between proven, probable, and possible in terms of therapy). Moreover, it would be useful to know how you managed those IFIs.

R: We thank the reviewer for the comment. We modified the manuscript and we clarified how we managed different IFIs. There were no difference between possible, probable and proven IFI in terms of therapy. (page 3, line 132-146)

I suggest reading and discuss these published papers that treat the same subject: 10.3390/idr14030041, 10.3892/br.2022.1517

R: We thank the reviewer for the suggestion.